# A large-scale transgenic RNAi screen identifies transcription factors that modulate myofiber size in *Drosophila*

**Flavia A. Graca**[1], **Natalie Sheffield**[1], **Melissa Puppa**[1¤], **David Finkelstein**[2], **Liam C. Hunt**[1], **Fabio Demontis**[1]*

**1** Department of Developmental Neurobiology, St. Jude Children's Research Hospital, Memphis, Tennessee, United States of America, **2** Department of Computational Biology, St. Jude Children's Research Hospital, Memphis, Tennessee, United States of America

¤ Current address: College of Health Sciences, University of Memphis, Memphis, Tennessee, United States of America

* Fabio.Demontis@stjude.org

**Data Availability Statement:** The authors affirm that all data necessary for confirming the conclusions of the article are present within the article, figures, and supplemental tables. The RNAi

## Abstract

Myofiber atrophy occurs with aging and in many diseases but the underlying mechanisms are incompletely understood. Here, we have used >1,100 muscle-targeted RNAi interventions to comprehensively assess the function of 447 transcription factors in the developmental growth of body wall skeletal muscles in *Drosophila*. This screen identifies new regulators of myofiber atrophy and hypertrophy, including the transcription factor Deaf1. Deaf1 RNAi increases myofiber size whereas Deaf1 overexpression induces atrophy. Consistent with its annotation as a Gsk3 phosphorylation substrate, Deaf1 and Gsk3 induce largely overlapping transcriptional changes that are opposed by Deaf1 RNAi. The top category of Deaf1-regulated genes consists of glycolytic enzymes, which are suppressed by Deaf1 and Gsk3 but are upregulated by Deaf1 RNAi. Similar to Deaf1 and Gsk3 overexpression, RNAi for glycolytic enzymes reduces myofiber growth. Altogether, this study defines the repertoire of transcription factors that regulate developmental myofiber growth and the role of Gsk3/Deaf1/glycolysis in this process.

## Author summary

Several diseases in humans reduce skeletal muscle mass and such wasting contributes to poor prognosis. Muscle mass is modulated in adulthood primarily by changes in the size of composing muscle cells (myofibers). Many of the signaling pathways that modulate myofiber size impinge on transcription factors. However, only few of the ~1,400 human transcription factors have been studied for their capacity to modulate skeletal muscle mass. To start to fill this gap in knowledge, we have used *Drosophila melanogaster* for testing the function of evolutionary conserved transcription factors. Because of the reduced genetic redundancy and the 40-fold increase in muscle mass that occurs in development, larval body wall skeletal muscles provide an ideal setting for identifying interventions that induce myofiber atrophy and hypertrophy. Here, we report the phenotype of >1,100

screen data is provided in S1 Table, the RNA-seq data is provided in S2 Table, and the phenotypes of RNAi lines for glycolytic enzymes are reported in S3 Table. Additional primary data, relative to Figs 2 and 3, are reported in S4 Table.

**Funding:** This work was supported by research grants to F.D. from the National Institute on Aging of the NIH (R01AG055532 and R56AG063806), the American Federation for Aging Research, the Ellison Medical Foundation (New Scholar in Aging award), the American Parkinson Disease Association, the Glenn Foundation for Medical Research, the Hartwell Foundation (Individual Biomedical Research award), and the American Lebanese Syrian Associated Charities. The funders had no role in study design, data collection and analysis, decision to publish, or preparation of the manuscript.

**Competing interests:** The authors have declared that no competing interests exist.

muscle-targeted RNAi interventions in myofiber size regulation. These include the transcription factor Deaf1, which induces hypertrophy upon RNAi and atrophy upon overexpression. These effects stem from modulation of glycolysis by Deaf1. Moreover, glycolysis is also modulated by the kinase Gsk3, which induces myofiber atrophy and muscle transcriptional changes similar to Deaf1. Altogether, our study provides a resource for understanding the function of 447 transcription factors in myofiber size regulation.

## Introduction

Skeletal muscle is a key tissue of the human body accounting for approximately 40–50% of the total body mass. A balance between muscle protein synthesis and breakdown is essential for maintaining the functionality and size of skeletal muscles [1]. When muscle protein synthesis exceeds protein degradation, this leads to skeletal muscle hypertrophy, which typically results from an increase in myofiber size. Conversely, myofiber atrophy occurs when protein breakdown is excessive or protein synthesis is insufficient [1,2]. This occurs following inactivity, fasting, as a side effect of many pharmacological treatments, and in the course of many degenerative diseases such as cancer cachexia, chronic heart disease, diabetes, sepsis, infections, chronic obstructive pulmonary disease, and renal failure [3]. Importantly, the loss of muscle mass is not just a side-effect of these conditions but a rather important contributor to morbidity and mortality. Strikingly, prevention of skeletal muscle mass loss in tumor-bearing mice results in increased survival even if cancer progression is not halted [4–6]. Despite great strides towards understanding the mechanisms responsible for muscle wasting, incomplete knowledge in this area has hampered the development of suitable therapies.

Gene expression changes are fundamental drivers of myofiber atrophy [7]. Many signaling pathways that induce atrophy impinge on key transcription factors to promote muscle protein degradation [1,8–10]. For example, forkhead box O (FoxO) transcription factors are activated in response to decreased insulin/IGF signaling and induce the expression of components of the autophagy-lysosome and ubiquitin-proteasome systems which in turn mediate protein degradation [11–17]. However, apart from a few transcription factors that have been extensively studied [1,8], much remains to be learnt on the role that the ~1,400 transcription factors encoded by the human genome [18] play in skeletal muscle mass homeostasis.

*Drosophila* body wall skeletal muscles have emerged as an important model system to determine the mechanisms of muscle growth and differentiation [19–34]. Previously, we have found that FoxO overexpression in larval body wall skeletal muscles leads to myofiber atrophy and reduces developmental muscle growth [35], suggesting that the fruit fly *Drosophila melanogaster* can be used to identify evolutionary-conserved regulators of myofiber size [10,36,37]. Here, by examining the impact of transgenic RNAi on developmental muscle growth, we have tested 1,114 RNAi lines targeting 447 of the 708 transcription factors encoded by the *Drosophila* genome [38]. Our study provides information on many novel transcription factors necessary for myofiber size determination. These include the transcription factor Deaf1 that is annotated as a phosphorylation target of the kinase GSK3-b [39], which is a known inducer of atrophy [40,41]. Similar to Gsk3, Deaf1 overexpression induces myofiber atrophy, whereas Deaf1 RNAi induces myofiber hypertrophy. Gene expression profiling further indicates that Gsk3/Deaf1-induced changes in myofiber size are associated with corresponding changes in the expression of glycolytic enzymes. Altogether, this study expands the repertoire of transcription factors that are implicated in myofiber size determination and indicates a possible role of Deaf1 in this process.

## Results

### A RNAi screen targeting transcription factors in *Drosophila* body wall skeletal muscles identifies regulators of muscle atrophy and hypertrophy

Previously, it was found that novel regulators of myofiber size can be uncovered by testing their function in *Drosophila* body wall larval skeletal muscles. For example, the vast increase in muscle size (~40-fold) that occurs over ~5 days of *Drosophila* larval development is modulated by the insulin/Akt/TOR signaling pathway [35], which is a well-known modulator of atrophy and hypertrophy in mammals [1]. Moreover, loss of UBR4, a ubiquitin ligase implicated in atrophy-associated muscle proteolysis [42,43], induces hypertrophy in *Drosophila* and in mammals [37]. Altogether, these studies indicate that homologous signaling pathways modulate myofiber size in mammals and in the developing *Drosophila* larvae. Moreover, *Drosophila* body wall muscles constitute an excellent setup for the identification of transcription factors that regulate myofiber size, as found for FoxO and Mnt [35].

On this basis, we took advantage of the simplicity of this system and the availability of transgenic RNAi resources for tissue-specific modulation of gene function to interrogate the role of evolutionary-conserved transcription factors in myofiber size regulation. For these studies with UAS/Gal4, the skeletal muscle-specific *Mef2-Gal4* [44] was crossed with 1,114 transgenic RNAi lines (from the VDRC and Bloomington stock centers) to target 447 of the 708 transcription factors encoded by the *Drosophila* genome [38]. *Mef2-Gal4* drives transgene expression in the body wall musculature located beneath the epidermis, and composed of muscles with stereotypical sizes, each consisting of a single myofiber. Because skeletal muscle-specific interventions that regulate the size of body wall muscles correspondingly change the size of the larva [35,37], we have scored the size of larvae as a convenient readout to assess the outcome of muscle-specific RNAi interventions (Fig 1A).

Compared to control RNAi, ~88% RNAi interventions lead to 3<sup>rd</sup> instar larvae of normal size, indicating that these RNAi do not impact developmental muscle growth. However, there were RNAi interventions that lead to larval lethality (~1.6%) and various degrees of atrophy (~3.7%), indicating that transcription factors targeted by these RNAi are necessary for optimal skeletal muscle growth. Conversely, ~3% of RNAi lead to hypertrophy, indicating that the transcription factors targeted by these RNAi normally limit muscle growth. Additionally, there were ~3% of RNAi interventions that rather than affecting size primarily affected larval shape, leading to thin or sickle-shaped larvae (Fig 1B and S1 Table).

RNAi interventions that induce atrophy at the larval stage typically do not develop into adult flies [35]. Therefore, it is not surprising to find that ~16% of *Mef2-Gal4*-driven RNAi interventions did not yield any adults, as these include RNAi interventions that impact larval stages of muscle growth as well as pupal stages of muscle remodeling. However, there were some muscle-related phenotypes that were manifested in adult flies obtained from other RNAi crosses. These included early lethality of adult flies soon after eclosion, as well as defects in wing position. Normally, the wings are kept at stereotypical positions in adult flies but developmental defects that cause muscle degeneration lead to upheld and/or depressed wings, as found before for pink1/parkin loss [45] and here for ~1% of RNAi interventions that target transcription factors (Fig 1C and S1 Table). Altogether, by using muscle-restricted RNAi screening, we have here identified novel transcription factors that impact developmental skeletal muscle growth.

Among the many screen hits identified for their capacity to regulate myofiber size, there were 17 genes that scored consistently with 2 or more RNAi lines and that therefore are more likely to be robust regulators of muscle growth (S1 Table). To further test these genes, we re-screened them with *Mef2-Gal4* and also with an additional muscle-specific driver, *MhcK-Gal4*,

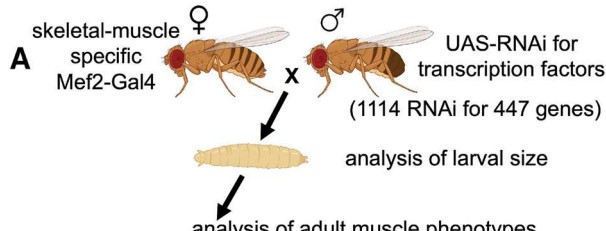

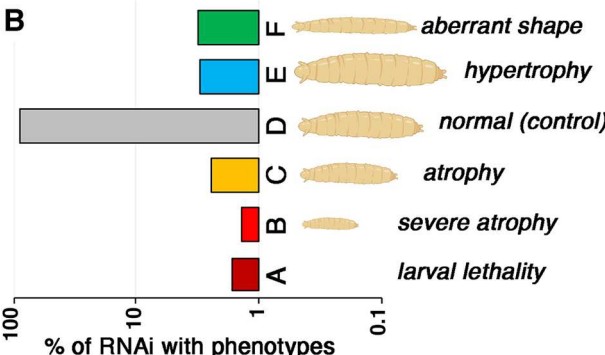

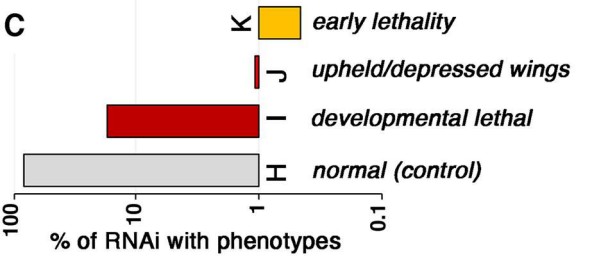

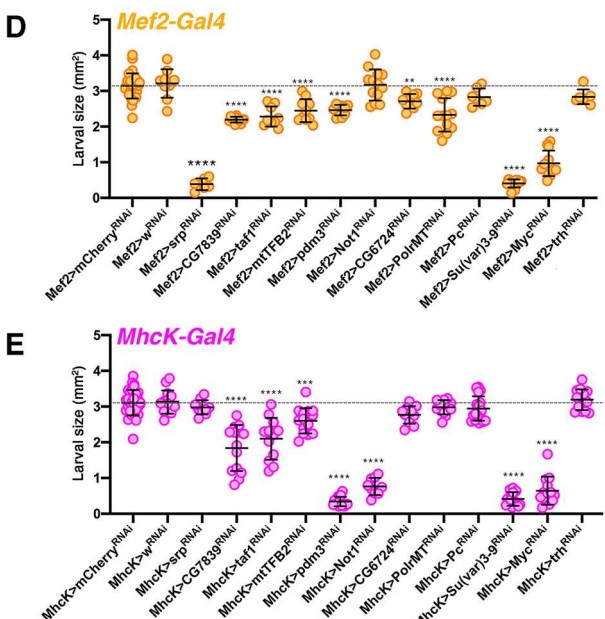

**Fig 1. Muscle-specific RNAi screening identifies transcription factors that modulate developmental growth of *Drosophila* body wall skeletal muscles.** (A) Scheme for the identification of transcription factors that modulate developmental skeletal muscle growth in *Drosophila*. The skeletal muscle-specific *Mef2-Gal4* was crossed with a collection of 1114 transgenic RNAi lines that target 447 transcription factors and transcriptional regulators to test their function in developmental muscle growth. *Mef2-Gal4* drives transgene expression in the body wall musculature located beneath the epidermis, and composed of muscles with stereotypical sizes, each consisting of a single myofiber. Because skeletal muscle-specific interventions that regulate the size of body wall muscles correspondingly change the size of the larva [35,37], we have scored the size of larvae as convenient readout to assess the outcome of muscle-specific RNAi interventions. Moreover, in cases where adult flies eclosed, also wing positioning was scored as upheld or depressed wings can indicate muscle developmental defects and degeneration. (B) Compared to control RNAi, most RNAi interventions lead to 3rd instar larvae of normal size, indicating that these RNAi interventions do not impact developmental muscle growth. There were RNAi interventions that lead to larval lethality and various degrees of atrophy, indicating that transcription factors targeted by these RNAi are necessary for optimal skeletal muscle growth. Conversely, RNAi for another subset of transcriptional regulators lead to hypertrophy, indicating that the transcription factors targeted by these RNAi interventions normally limit muscle growth. Additionally, there were certain RNAi intervention that rather than affecting size primarily affected the shape of the larva, leading to thin or sickle-shaped larvae. (C) Although RNAi interventions that induce atrophy at the larval stage do not develop into adult flies, we have examined the adults that eclosed from all other RNAi interventions. The wings are kept at stereotypical positions in adult flies but developmental defects leading to muscle degeneration are known to lead to upheld and/or depressed wings, as found here for RNAi of several transcription factors. A full report of screen results is shown in S1 Table. (D-E) The area of larvae with muscle-specific expression of transgenic RNAi driven by *Mef2-Gal4* (D) and by *MhcK-Gal4* (E). Similar results are obtained with RNAi driven by both drivers, although some differences are found in the phenotypes induced by *Mef2-Gal4* versus *MhcK-Gal4*, presumably because of differences in the potency and tissue-specificity of these Gal4 lines. Mean±SD and N = 6–31 (D) and N = 9–32 (E) are shown; **$P<0.01$, ***$P<0.001$, ****$P<0.0001$. The scheme in Fig 1 was drawn with BioRender.

which drives strong transgene expression from the embryonic stage of muscle development [46]. However, different from *Mef2-Gal4*, *InR* overexpression with *MhcK-Gal4* only marginally increased larval size (S1 Fig), suggesting that *MhcK-Gal4* might not be suitable to uncover muscle hypertrophy phenotypes compared to *Mef2-Gal4*. On this basis, we have re-screened with *MhcK-Gal4* only RNAi lines (from the Bloomington collection) with which we previously obtained a reduction in body size upon muscle-specific expression.

This follow-up analysis revealed that larval size is overall similarly affected by RNAi driven by *Mef2-Gal4* (Fig 1D) and by *MhcK-Gal4* (Fig 1E). However, some discrepancies in the phenotypes induced by *Mef2-Gal4* versus *MhcK-Gal4* were also observed, presumably because of differences in the potency and tissue-specificity of these Gal4 lines (Figs 1D–1E and S2). Specifically, RNAi for CG7839, taf1, mtTFB2, pdm3, Su(var)3-9, and myc (dm) consistently reduced larval body area with both *Mef2-Gal4* and *MhcK-Gal4*, although pdm3 RNAi yielded stronger effects with *MhcK-Gal4* versus *Mef2-Gal4*. However, RNAi for srp and PolrMT significantly reduced larval size with *Mef2-Gal4* but not with *MhcK-Gal4*. Moreover, RNAi for CG6724 marginally reduced larval size with both *Mef2-Gal4* and *MhcK-Gal4*, although this was statistically significant only with *Mef2-Gal4*. Conversely, RNAi for Not1 reduced larval size only when driven by *MhcK-Gal4*. Lastly, RNAi for Pc and tracheales (trh), which were previously classified as screen hits, did not significantly impact larval size when rescreened with *Mef2-Gal4* (Fig 1D), suggesting that they are false positives. Altogether, despite some differences, RNAi driven by *Mef2-Gal4* and *MhcK-Gal4* similarly impacts myofiber size (Fig 1D and 1E), indicating that our screen strategy is appropriate for finding candidate regulators of muscle growth.

The small set of high-confidence regulators of muscle growth includes genes with functional homologs in humans, i.e. CG7839/CEBPZ, taf1/TAF1, the mitochondrial transcription factor mtTFB2/TFB2M, pdm3/POU6F2, Su(var)3-9/SUV39H, and dm/MYC (Fig 1D and 1E and S1 Table). Although these transcription factors have not been previously implicated in muscle growth or wasting, apart dm/MYC [35], some were found to modulate muscle differentiation in mice. Specifically, TAF1 has been previously implicated in myogenesis via its

capacity to bind Pax3 and modulate its ubiquitination and proteasomal degradation [47] whereas the histone methyltransferase SUV39H1 was found to repress MyoD-stimulated myogenic differentiation [48].

Altogether, these findings indicate that RNAi screening in *Drosophila* is a useful approach to identify novel candidate regulators of myofiber size determination.

## Muscle-specific RNAi for screen hits identifies transcription factors that modulate developmental myofiber growth in *Drosophila*

We have conducted a large-scale RNAi screen for transcription factors that regulate developmental skeletal muscle growth in *Drosophila*. Because body wall skeletal muscles are located beneath the epidermis, genetic interventions that regulate muscle size correspondingly change the size of the larva [35,37]. Assessing larval size is an easily-scorable screen readout that has led to the identification of many regulators of developmental muscle growth (Fig 1).

To better test the impact of screen hits, we have determined their impact on myofiber size via larval dissections and analysis of body wall skeletal muscles. Specifically, the outcome of some muscle-specific interventions that affected larval size was validated via the analysis of a set of representative muscles, ventral longitudinal VL3 and VL4 muscles, which are each composed by a single myofiber with a stereotypical size [35,37].

For these studies, we selected a set of genes based on their extremely high evolutionary conservation (i.e., typically, a DIOPT score >7), consistent scoring with multiple RNAI lines, and/or novelty (Fig 2 and S1 Table). Compared to controls (white$^{RNAi}$ and mcherry$^{RNAi}$), RNAi for screen hits driven in skeletal muscle by *Mef2-Gal4* led to decreased (atrophy) and increased (hypertrophy) size of VL3 and VL4 skeletal muscles. Quantification of the cumulative area of VL3 and VL4 muscles from multiple larvae indicates that RNAi for Nurf-38, e(y)1, alien, CG7839, Taf1, MBD-R2, mtTFB2, pdm3, and dati reduces VL3+VL4 muscle area (atrophy). This indicates that these transcription factors are necessary for optimal myofiber growth during larval development. Conversely, muscle-specific RNAi for FoxO, Cnc, and Deaf1 increases the area of VL3+VL4 muscles (hypertrophy), indicating that these transcription factors normally limit developmental myofiber growth. Altogether, these histological analyses confirm that this muscle-targeted RNAi screen has identified novel transcription factors that regulate myofiber developmental growth.

## Deaf1 RNAi induces myofiber hypertrophy whereas Deaf1 overexpression causes myofiber atrophy

Among the many regulators of skeletal muscle homeostasis identified in this screen, RNAi interventions that induce myofiber hypertrophy are the most interesting as they highlight transcription factors that normally impede growth and that could be inhibited to contrast wasting. Among them, the transcription factor Deaf1 (deformed epidermal autoregulatory factor 1) has been previously implicated in early development and innate immunity in *Drosophila* [49–52] and in human neurodevelopmental disorders [53–55] but not in muscle growth. On the basis of this possible novel function of Deaf1 in muscle, we further examined its impact on myofiber size determination. Whereas Deaf1 RNAi induces hypertrophy compared to control RNAi (Fig 2D), muscle-restricted Deaf1 overexpression led to myofiber atrophy (Fig 3A). Specifically, the area of ventral longitudinal VL3 and VL4 muscles is lower upon *Deaf1* overexpression (Fig 3B), whereas it increases upon *insulin receptor* (*InR*) overexpression, as expected [35].

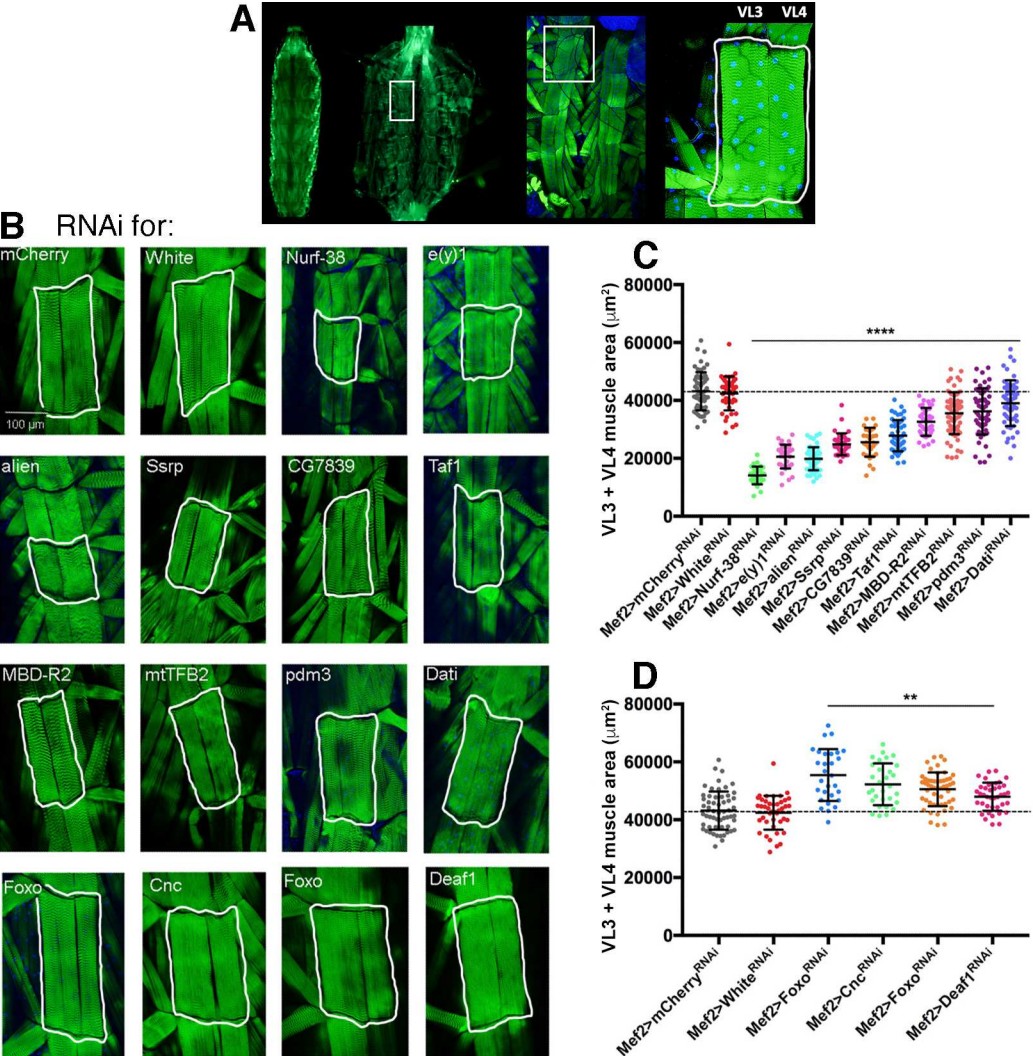

**Fig 2. Muscle-specific RNAi for screen hits identifies transcription factors that modulate developmental myofiber growth in *Drosophila*.** (A) Validation of RNAi screen hits via dissection of 3[rd] instar larvae and confocal imaging of ventral longitudinal VL3 and VL4 body wall skeletal muscles, which have stereotypical sizes. (B) Compared to controls (white[RNAi] and mcherry[RNAi]), RNAi for screen hits driven in skeletal muscle by *Mef2-Gal4* leads to a decrease (atrophy) and an increase (hypertrophy) in the size of VL3 and VL4 skeletal muscles, each consisting of a single myofiber. (C) Quantitation of the cumulative area of VL3 and VL4 muscles from multiple larvae indicates that RNAi for Nurf-38, e(y)1, alien, CG7839, Taf1, MBD-R2, mtTFB2, pdm3, and Dati reduces VL3+VL4 muscle area (atrophy). This indicates that these transcription factors are necessary for optimal myofiber growth during larval development. (D) Conversely, muscle-specific RNAi for Foxo, Cnc, and Deaf1 increases the area of VL3+VL4 muscles (hypertrophy), indicating that these transcription factors limit developmental myofiber growth. N = 12–70 and mean±SD is shown; **$P<0.01$, ***$P<0.001$.

## Gsk3 regulates myofiber size similar to Deaf1

Although Deaf1 has not been implicated in myofiber size determination, it was previously identified as a phosphorylation target of the glycogen synthase kinase GSK3 [39], which induces myofiber atrophy and muscle wasting in response to many catabolic stimuli via the phosphorylation of target proteins in mice [14,41,56]. Specifically, it was found that DEAF1 interacts with and is phosphorylated by GSK3A and GSK3B [39]. To determine whether shaggy (the Drosophila homolog of GSK3 and GSK3B) regulates myofiber size in Drosophila as found in mammals, we modulated its activity via overexpression of constitutive active

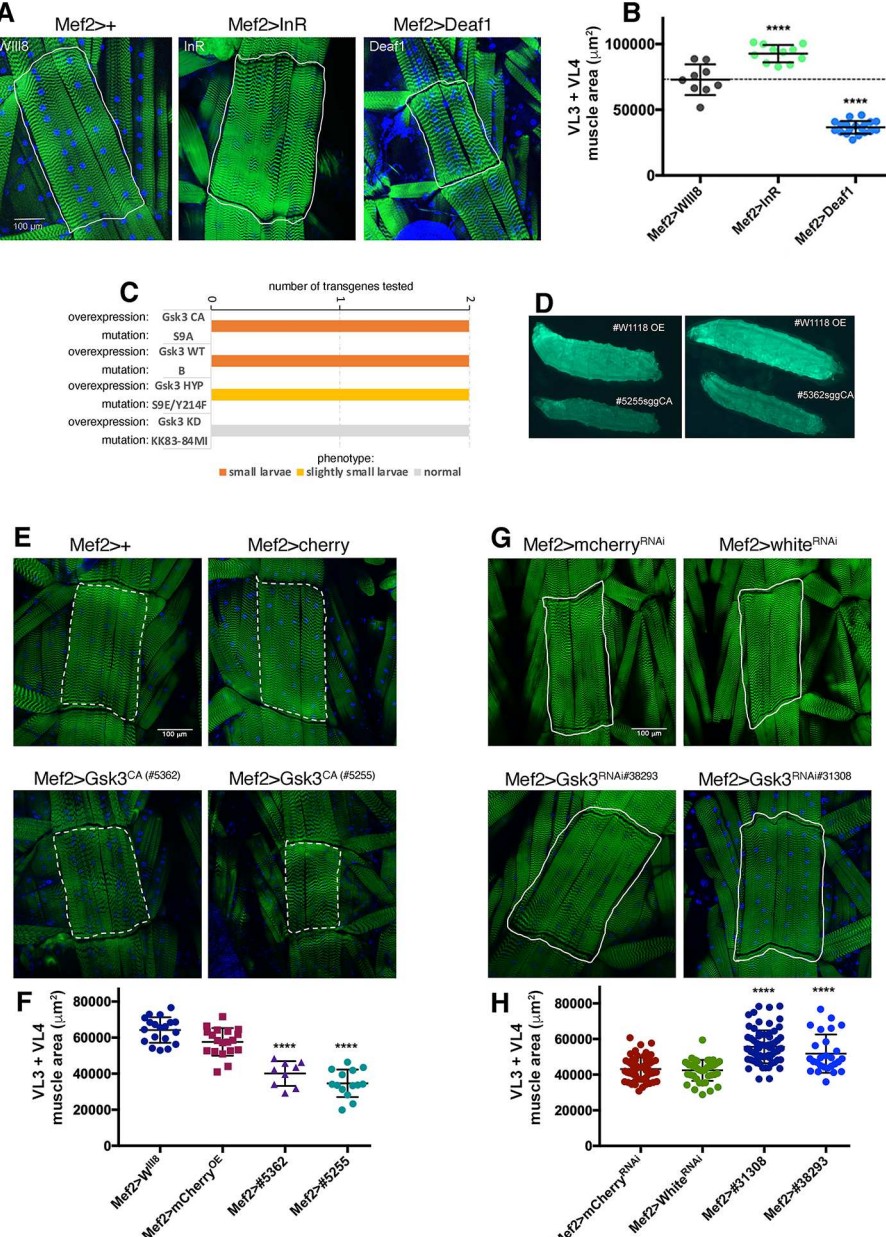

**Fig 3. Muscle-restricted activity of Deaf1 and Gsk3 impedes developmental muscle growth.** (A) Transgenic overexpression of the transcription factor *Deaf1*, driven specifically in skeletal muscle with *Mef2-Gal4*, leads to a reduction in the area of larval muscles, each consisting of a single myofiber, compared to controls with no transgene expression (*Mef2>+*). (B) The area of ventral longitudinal VL3 and VL4 muscles is lower upon *Deaf1* overexpression, whereas it increases upon *insulin receptor* (*InR*) overexpression. N = 9–19 and mean±SD is shown; ****$P<0.001$. (C) Gsk3 transgenic overexpression driven in skeletal muscle with Mef2-Gal4 leads to small size of 3rd instar larvae in a manner proportional to Gsk3 kinase activity: Gsk3 CA (constitutive active) and WT (wild-type) induce atrophy, hypomorphic (HYP) mutants with only limited kinase activity have little impact, whereas KD (kinase-dead) Gsk3 mutants do not affect developmental muscle growth. (D) Gsk3 activity in muscle reduces body size, as indicated by larvae where body wall skeletal muscle are shown by expression of Mhc-GFP. (E) Compared to controls (no transgene and *mcherry* overexpression), overexpression of 2 different *Gsk3*$^{CA}$ transgenes with *Mef2-Gal4* leads to a decrease (atrophy) in the size of myofibers, as indicated by the quantitation of the cumulative area of VL3 and VL4 muscles from multiple larvae (F); N = 9–19 and mean±SD is shown; ****$P<0.0001$. (G) Conversely, Gsk3 RNAi in skeletal muscle induces myofiber hypertrophy, as indicated by the the cumulative area of VL3 and VL4 muscles (H); N = 27–71 and mean±SD is shown; ****$P<0.0001$.

versions. Specifically, we employed *Mef2-Gal4* to drive expression of transgenes encoding for constitutive-active, wild-type, and kinase-dead Gsk3. As expected based on its function in higher organisms, we found that muscle-specific overexpression of constitutive active (CA) and wild-type (WT) *Gsk3* led to reduced body size, which is indicative of muscle atrophy (Fig 3C and 3D). On the other hand, Gsk3 variants with reduced kinase activity minimally impacted larval size, and kinase-dead Gsk3 mutants had no effect (Fig 3C and S3 Table).

To corroborate these findings, larvae with skeletal muscle-specific *Gsk3*$^{CA}$ overexpression were dissected and the area of ventral longitudinal VL3 and VL4 muscles analyzed. As expected based on the function of Gsk3 in mammalian systems [14,41,56] and our preliminary analyses (Fig 3C and 3D), Gsk3$^{CA}$ lead to significant decline in myofiber size (Fig 3E and 3F). To further assess the role that Gsk3 plays in *Drosophila* body wall muscle growth, we reduced its levels in skeletal muscle via RNAi. Conversely to Gsk3 activation (Fig 3C–3F), Gsk3 RNAi induced myofiber hypertrophy, compared to control RNAi lines against *white* and *mcherry* (Fig 3G and 3H).

Altogether, these findings indicate that Gsk3, similar to Deaf1, regulates muscle mass in *Drosophila* (Fig 3), as observed in mammals [14,41,56]. Because Deaf1 was found to be a phosphorylation target of Gsk3 [39], these findings suggest that Deaf1 may negatively regulate myofiber size in *Drosophila* skeletal muscle by acting downstream of Gsk3 signaling.

## Gsk3 and Deaf1 induce similar gene expression changes in *Drosophila* body wall skeletal muscles

Because the transcription factor DEAF1 has been previously reported to interact with and to be phosphorylated by GSK3A/B [39], and it similarly regulates myofiber size (Fig 3), we next examined whether Gsk3 and Deaf1 induce similar gene expression changes in *Drosophila* body wall skeletal muscle. For these studies, we used *Mef2-Gal4* to modulate the levels of Deaf1 and Gsk3 in muscle. As expected, Deaf1 mRNA levels were significantly lower upon Deaf1 RNAi and higher upon Deaf1 overexpression, respectively. Similarly, higher Gsk3 levels were found upon its overexpression (Fig 4A).

RNA sequencing from filleted larvae (which consist primarily of body wall skeletal muscles and the associated epidermis) identified many transcriptional changes that occur upon Deaf1 RNAi in comparison to control white RNAi. Cross-comparison with gene expression changes induced by *Deaf1* overexpression revealed that significantly regulated genes (p<0.05) are largely regulated in opposite fashions by Deaf1 RNAi and Deaf1 overexpression ($R^2$ = 0.46), each normalized to its respective control (Fig 4B). Moreover, comparison of the muscle transcriptomes revealed that gene expression changes induced by *Deaf1* overexpression are highly overlapping ($R^2$ = 0.49) with those induced by constitutive active Gsk3 (Fig 4C).

GO term analysis of categories enriched among genes upregulated by Deaf1 RNAi indicates that Deaf1 RNAi may induce myofiber hypertrophy by promoting glycolysis, sarcomere organization, and by modulating the function of histone deacetylases (Fig 4D). In particular, glycolysis represents the top category of genes upregulated by Deaf1 RNAi and, consistent with transcriptome cross-comparisons (Fig 4B and 4C), Deaf1 and Gsk3 overexpression induce converse changes, i.e. significantly reduce expression of most glycolytic enzymes (Fig 4E). Altogether, these findings suggest that the transcription factor Deaf1 may regulate myofiber size via the transcriptional modulation of several target genes, including glycolytic enzymes.

## Expression of glycolytic enzymes sustains growth of larval body wall skeletal muscles

We have found that Deaf1 RNAi, which induces myofiber hypertrophy, promotes the expression of glycolytic enzymes whereas Deaf1 overexpression, similar to Gsk3, reduces their

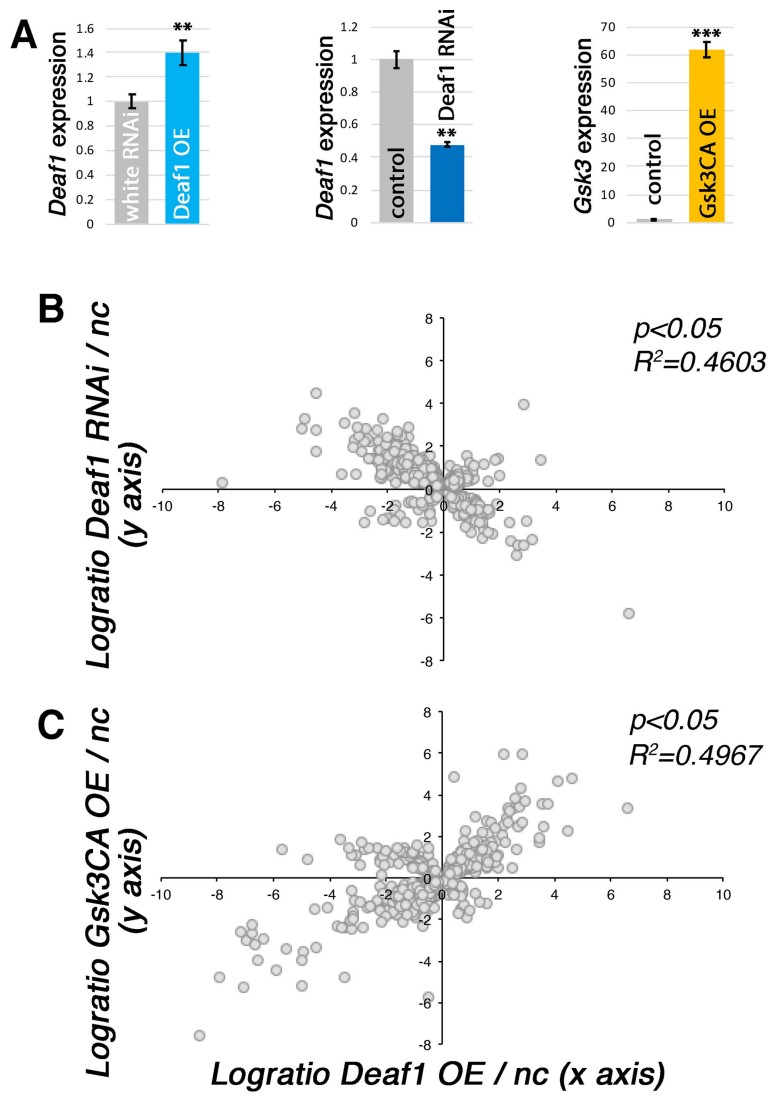

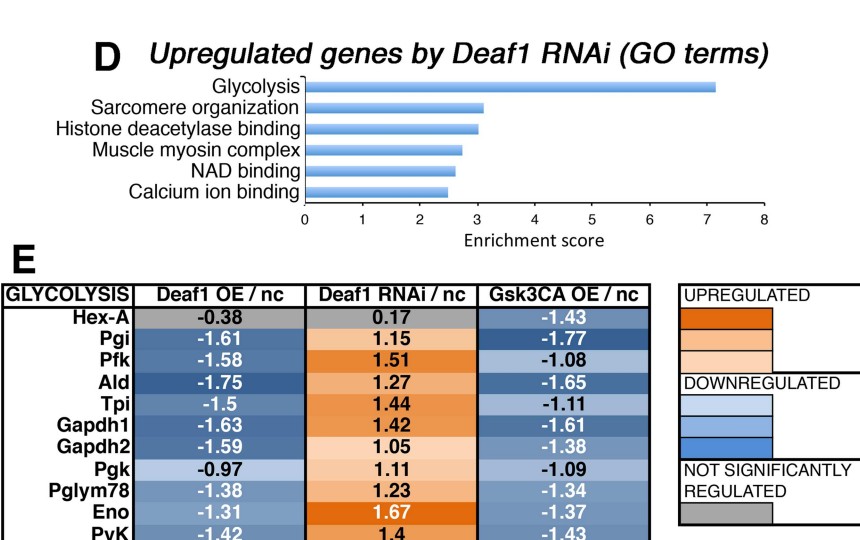

**Fig 4. GSK3 and Deaf1 induce similar gene expression changes in larval skeletal muscles.** (A) Validation of genetic interventions. Muscle-restricted *Deaf1* overexpression leads to an increase in Deaf1 mRNA levels, opposite to Deaf1 RNAi. As expected, GSK3 overexpression also increases Gsk3 mRNA levels. (B) Coincident with their opposite roles in regulating myofiber size, converse gene expression changes ($R^2 = 0.46$; genes regulated with $p<0.05$) are induced in larval body wall muscles by Deaf1 overexpression (OE) and Deaf RNAi, each normalized to their respective controls, i.e. no transgene overexpression and control white RNAi. (C) Largely similar gene expression changes are induced by overexpression of Deaf1 and of constitutive active (CA) Gsk3 ($R^2 = 0.49$; $p<0.05$). (D) DAVID GO term analysis reveals gene categories that are enriched among Deaf1-regulated genes, which include glycolysis. (E) Analysis of glycolytic genes reveals that most of them are significantly ($p<0.05$) and concordantly regulated by Deaf1 and Gsk3. Specifically, glycolytic enzymes are upregulated by Deaf1 RNAi compared to control whereas their expression is suppressed by Deaf1 OE and Gsk3CA OE. S2 Table reports the results of RNA sequencing.

expression (Fig 4). On this basis, we next tested the impact of glycolysis on skeletal muscle growth. To this purpose, we screened 44 RNAi lines targeting glycolytic enzymes with *Mef2-- Gal4* and found that many of them led to small larval size (Fig 5A and S3 Table), indicative of muscle atrophy [35]. We further analyzed some of the RNAi lines that target glycolytic enzymes. As expected, small body size due to expression of transgenic RNAi for *Eno* and *Pglym78* in skeletal muscle (Fig 5A) was associated with reduced size of VL3 and VL4 muscles (Fig 5B and 5C). These findings indicate that glycolysis is necessary for myofiber growth, in line with previous studies [57,58], and that it may indeed be a primary means by which Gsk3 and Deaf1 modulate myofiber growth in *Drosophila* body wall skeletal muscles.

## Discussion

In this study, we took advantage of a simple, genetically tractable, model organism, *Drosophila melanogaster* [59], to expand the knowledge about transcription factors that regulate myofiber size. Specifically, we have used transgenic RNAi to knock down the levels of evolutionary-conserved transcription factors in *Drosophila* larval body wall skeletal muscles which, by growing ~40-fold during few days of larval development, offer an ideal setup for identifying modulators of myofiber growth. Because these muscles are located right beneath and surround the larval epidermis, changes in muscle mass result in changes in the overall larval body size, making this system amenable for visual phenotypic screens [35,37]. Moreover, the reduced genetic redundancy of *Drosophila melanogaster* compared to mice and humans constitutes another advantage for uncovering regulators of myofiber size [22], as demonstrated by the lower number of transcription factors present in *Drosophila* compared to humans (708 versus ~1,400).

The RNAi screen here done provides insight into transcription factors that regulate myofiber growth. Because this screen included both DNA-binding transcription factors and transcriptional regulators that are part of larger nuclear complexes, we expect that the screen hits here identified may modulate myofiber size via a plethora of mechanisms. For example, the nucleosome remodeling factor Nurf-38 catalyzes ATP-dependent nucleosome sliding and facilitates transcription of chromatin [60] and this may constitute a mechanism by which it is necessary for myofiber growth. Another example is e(y)1/Taf9, i.e. TBP-associated factor 9, which encodes for a component of the transcription factor IID complex, which regulates transcription from core promoters and enhancer-promoter interactions [61] but also regulates lipid metabolism [62], which has been implicated in muscle wasting [63]. Overall, despite phenotypic similarity, the transcription factors and transcriptional regulators here identified may regulate myofiber size via distinct target genes and transcriptional/chromatin remodeling mechanisms.

The RNAi screen we have conducted has identified several candidate regulators of muscle growth. However, as found in other screens [64], some of these hits could be false positives. For example, *tracheless (trh)*, a transcription factor necessary for the development of the

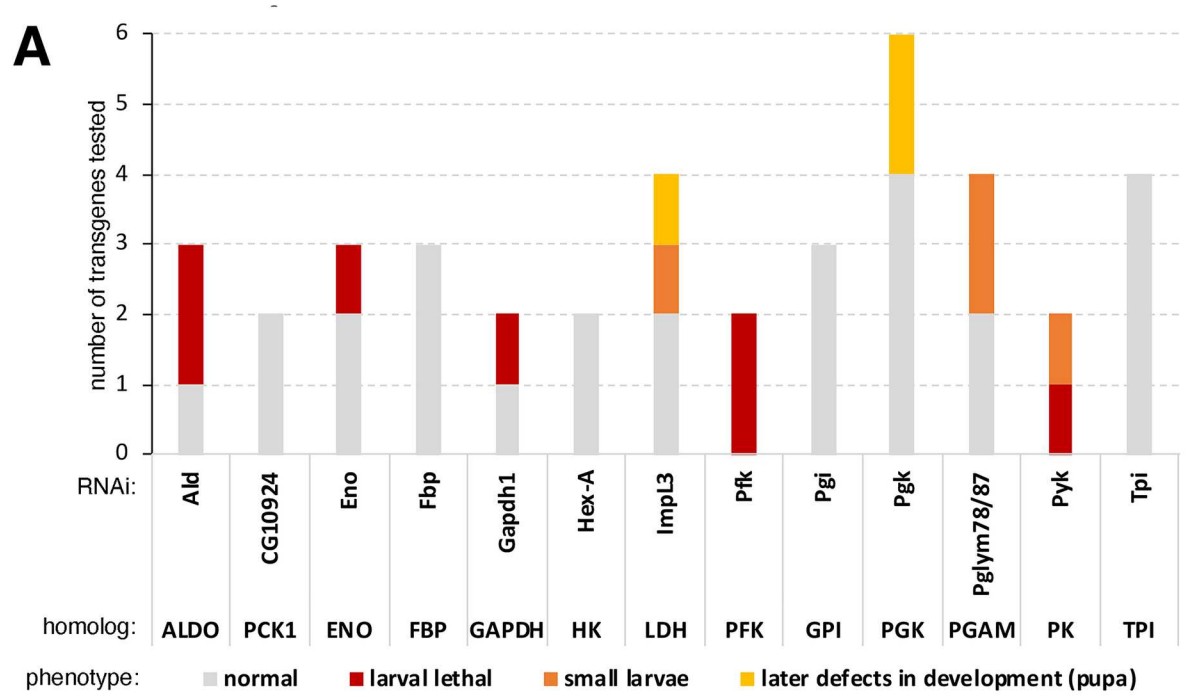

**Fig 5. RNAi for glycolytic enzymes impedes developmental growth of larval body wall skeletal muscles.** (A) RNAi for glycolytic enzymes driven in skeletal muscle with Mef2-Gal4 leads to small size of 3rd instar larvae, indicating that glycolysis is needed to sustain developmental muscle growth. A full list of phenotypes obtained with 44 lines targeting glycolysis is reported in S3 Table. (B) Transgenic RNAi for the glycolytic enzymes *Eno* and *Pglym78* reduces myofiber size, compared to control white$^{RNAi}$ and cherry$^{RNAi}$. (C) The area of ventral longitudinal VL3 and VL4 muscles, each consisting of a single myofiber, is reduced upon Eno$^{RNAi}$ and Pglym78$^{RNAi}$. N = 12–29 and mean±SD is shown; **$P<0.01$ and ***$P<0.001$.

*Drosophila* airway system [65], was initially identified as a screen hit but subsequent re-testing with *Mef2-Gal4* uncovered only minor phenotypes induced by trh RNAi (Fig 1D), suggesting that this is a false positive. While transcription factors that scored consistently with multiple RNAi lines (such as Deaf1 RNAi) are less likely to be false positives, screen hits identified with a single RNAi line most likely also consist in large part of bona fide muscle growth regulators (indeed in some cases only a single RNAi line was available to test the function of a given transcription factor).

Conversely, other transcription factors that are important regulators of muscle growth may not have scored because of technical limitations, i.e. they could be false negatives. Although the potency of transgenic RNAi depends on the Gal4 line used and specific RNAi collection, a previous study in the *Drosophila* embryo has found that phenotypes were observed only among the RNAi interventions that yielded a target gene knockdown greater than 50% [66]. Specifically, this study has found that, out of ~450 RNAi lines targeting kinases, ~29% were not functional (~12% did not display any knockdown, and an additional ~17% displayed a knockdown of less than 50% that did not yield a phenotype; [66]). Therefore, it is possible that around 1/3 of the lines tested in our screen is not functional and that therefore the muscle growth regulators targeted by these RNAi lines are not uncovered by our screen. On this basis, rather than being an exhaustive and definitive determination of the transcription factors that regulate muscle growth, our study provides a list of candidate regulators of myofiber size which should be further tested in *Drosophila* and other model organisms.

Another limitation of the screen consists in the tissue specificity of the *Mef2-Gal4* line that we have used. As originally described in ref. [35], this *Mef2-Gal4* line drives transgene expression in body wall skeletal muscles but also in visceral muscles and in some cells in the brain [35]. Closer analysis of such brain cells suggests that they consist of mushroom body neurons (S2 Fig), which have been found to express endogenous *Mef2* [67]. However, *Mef2-Gal4* fluorescence is most strongly observed in skeletal muscle compared to non-muscle cells and therefore it is unclear if sufficient target gene knockdown is achieved in non-muscle cells to generate a phenotype. Moreover, the overall similarity of phenotypes induced by transgenic RNAi driven by complementary muscle drivers (i.e. *Mef2-Gal4* and *MhcK-Gal4*) suggests that the observed muscle atrophy phenotypes are indeed due to RNAi expression in skeletal muscle (Fig 1D and 1E). Nonetheless, the changes in larval body size and myofiber growth observed in our study may in some cases depend on the modulation of the target gene outside of skeletal muscle. As in the case of false and negative screen hits, this constitutes a technical limitation of the study that will be resolved by complementary approaches for testing the function of the candidate regulators of myofiber size here identified.

Among the many screen hits identified, we have examined in more detail the function of Deaf1, an evolutionary conserved transcription factor that had not been previously implicated in skeletal muscle growth. Specifically, we have found that Deaf1 RNAi induces myofiber hypertrophy whereas Deaf1 overexpression causes atrophy. Mechanistically, Deaf1 RNAi promotes the expression of glycolytic enzymes whereas Deaf1 overexpression reduces it, suggesting that glycolysis is necessary for optimal skeletal muscle growth, as previously found in *Drosophila* [57] and in other contexts [68].

We also find that similar transcriptional responses, including expression of glycolytic enzymes, are induced by Deaf1 and Gsk3, a known inducer of myofiber atrophy [40,69–71] that interacts with and phosphorylates Deaf1 [39]. Many phosphorylation targets of GSK3 have been identified in mammals, including several transcription factors, such as MITF, NF-κB, and CREB [72–77]. However, much remains to understand about the GSK3 targets that are most necessary for GSK3 output in distinct tissues and disease conditions. In particular, although it is well established that GSK3 promotes muscle wasting [40,69–71], it is incompletely understood how GSK3 promotes transcriptional changes that drive muscle protein catabolism during atrophy. Our findings now suggest that Deaf1 may contribute at least in part to the transcriptional changes induced by Gsk3 activity in muscle, and that these may include the modulation of glycolysis.

Altogether, this study has expanded the repertoire of transcription factors that regulate myofiber growth and highlights a possible role for Gsk3, Deaf1, and glycolysis in this process.

## Materials and methods

### RNAi screening

The list of fly stocks used for RNAi screening is reported in S1 Table and refers to RNAi for *Drosophila* transcription factors that are evolutionarily conserved in humans, as defined based on a homology DIOPT [78] score of ≥2. For each screen cross, 10 *Mef2-Gal4* virgin females were crossed with 5 males for each RNAi line tested. Progenies were reared at 25˚C and transferred to new food every 4 days. Subsequently, the size of 3[rd] instar wandering larvae was scored in comparison with negative controls (white[RNAi]) and positive controls, which consisted in *FoxO* overexpression (which induces atrophy) and overexpression of *insulin receptor* (InR, which induces hypertrophy), as previously shown [35,37]. Specifically, larval size phenotypes due to *Mef2-Gal4*-driven RNAi in muscle were scored as follows: A- no larvae/no pupae; B- very small larvae/no pupae (i.e. smaller than *FoxO* overexpression); C- small larvae/small pupae (i.e. similar to *FoxO* overexpression); D- normal (no visible phenotype); E- increased larval/pupal size (i.e. similar to *InR* overexpression); and F- thin or sickle-shaped larvae with locomotor defects.

Adult flies obtained from RNAi crosses were scored based on the following categories: H-normal; I- no adults eclosed, i.e. developmental lethal; J- upheld or depressed wings in the majority of flies in the tube (i.e. similar to pink1 RNAi or parkin RNAi); and K- early lethality of eclosed flies. Normal muscle development results in stereotypical wing positioning, which is present in white[RNAi] control flies, whereas upheld/depressed wings are an indication of improper muscle development and/or muscle degeneration [45]. RNAi interventions that lead to small larvae and pupae (A-C) and larvae with aberrant shape (F) typically do not give rise to adult flies, as previously shown [35].

### *Drosophila* stocks

In addition to *Mef2-Gal4* [44], the following fly stocks were used: UAS-Deaf1 [50], UAS-foxo and UAS-InR [35,79], and stocks for overexpression of wild-type, constitutive-active, and kinase-dead Gsk3 transgenes [80], which are reported in S3 Table. MhcK-Gal4 (Mhc-GAL4.K, BL#55133; [46]) was used for studies in Fig 1E. The list of fly stocks used for RNAi screening is reported in S1 Table whereas the list of RNAi lines that target glycolytic enzymes is reported in S3 Table.

The fly stocks utilized for VL3+VL4 muscle analyses (Fig 2) are the following: UAS-white-[RNAi] (BL#33623), UAS-cherry[RNAi] (BL#35785), UAS-Nurf-38[RNAi] (BL#31341), UAS-e(y)1[RNAi] (BL#32345), UAS-alien[RNAi] (BL#28908), UAS-Ssrp[RNAi] (BL#26222), UAS-CG7839[RNAi]

(BL#25992), UAS-Taf1$^{RNAi}$ (BL#32421), UAS-MBD-R2$^{RNAi}$ (BL#27029), UAS-mtTFB2$^{RNAi}$ (BL#27055), UAS-pdm3$^{RNAi}$ (BL#35726), UAS-dati$^{RNAi}$ (BL#26711), UAS-foxo$^{RNAi}$ (BL#27656 and BL#32993), UAS-Cnc$^{RNAi}$ (BL#32863), and UAS-Deaf1$^{RNAi}$ (BL#32512).

The fly stocks utilized for RNA-seq (Fig 4) are the following: UAS-Deaf1$^{RNAi}$ (BL#32512), UAS-white$^{RNAi}$ (BL#33623), UAS-Deaf1 [50], UAS-Gsk3$^{CA}$ (BL#5255), and w$^{1118}$ (+).

## Staining of body wall skeletal muscles, confocal microscopy, and image analysis

Male larvae were dissected in ice-cold Ca$^{2+}$-free, MgCl$_2$-free PBS (Gibco) and filleted larval samples were fixed for 20 minutes in PBS with 4% paraformaldehyde, as previously done [35]. After washing with PBS, body wall skeletal muscles were stained overnight with DAPI (4',6-diamidino-2phenylindole, 1μg/mL) to visualize nuclei, and imaged to detect the endogenous fluorescence of a Mhc-GFP fusion protein. Body wall muscles were mounted on microscope slides and the VL3/4 muscles were imaged using a Zeiss LSM880 confocal laser-scanning microscope. Confocal images were analyzed using the measure tools of the ImageJ software to quantitate the area of VL3+VL4 muscles for each sample. Larval body size was quantified with ImageJ.

## RNA sequencing

RNA-seq was done following similar procedures as before [81,82]. Specifically, total RNA was extracted from filleted *Drosophila* larvae, which consist primarily of body wall skeletal muscles. Three biological replicates were prepared for RNA-seq with the TruSeq stranded mRNA library preparation kit (Illumina) and sequenced on the Illumina HiSeq 4000 platform, with six samples in each lane. Multiplexing was done on a per flowcell basis. Approximately 100 million reads were obtained for each sample. FASTQ sequences derived from mRNA paired-end 100-bp sequences were mapped to the *Drosophila melanogaster* genome (BDGP5) with the STAR aligner (version 2.5.3a) [83]. Transcript level data were counted using HTSeq (version 0.6.1p1) [84] based on the BDGP5 GTF release 75. The TMM method [85] was used to calculate the normalization factors. Then, linear modeling was carried out on the log2(CPM) (count per million) values where the mean-variance relationship is accommodated using precision weights calculated by the voom function [86] of the limma package in R 3.2.3 (R Core Team 2013; [87]). A q-value (FDR) was calculated for multiple comparison adjustments of RNA-seq data. The lmFit, eBayes, and contrasts.fit functions from the limma package were used for the linear modeling. Statistical analyses were performed using log2(FPKM) values in Partek Genomic Suite 6.6 (www.partek.com/partek-genomics-suite/). The gene sets were analyzed by DAVID (Database for Annotation Visualization and Integrated Discovery) to identify enriched functional classes of genes [88].

The RNA-seq comparisons refer to *Mef2>Deaf1RNAi(BL#32512)* versus *Mef2>whiteRNAi (BL#33623)*, *Mef2>Deaf1* versus *Mef2>+*, and *Mef2>Gsk3CA(BL#5255)* versus *Mef2>+*; (n = 3/genotype). The RNA-seq data is reported in S2 Table and has been deposited to the Gene Expression Omnibus with accession number GSE174637.

## Statistical analysis

All data points refer to biological replicates and the number is indicated in the figure legends. Each biological replicate refers to data obtained from a different larva; typically, larvae from 2 or more crosses were analyzed for each genotype. The larvae analyzed in Figs 2–5 were obtained from crosses different from those used for the screen. The unpaired two-tailed Student's t-test was used to compare the means of two independent groups to each other. One-

way ANOVA with Tukey's post hoc test was used for multiple comparisons of more than two groups of normally distributed data. Bar graphs present the mean ± SEM or ± SD, as indicated in the figure legends. Throughout the figures, asterisks indicate the significance of the p value: $^*p<0.05$; $^{**}p<0.01$; $^{***}p<0.001$. A significant result was defined as $p<0.05$. Statistical analyses were done with Excel and GraphPad Prism.

## Supporting information

**S1 Fig. Comparison of *Insulin Receptor* overexpression with *Mef2-Gal4* and *MhcK-Gal4*.** Consistent with previous studies in *Drosophila* and mammals, overexpression of *insulin/IGF receptor* (*InR*) in skeletal muscle via *Mef2-Gal4* induces skeletal muscle hypertrophy, as indicated by the increase in body size. However, a relatively minor increase is found with *InR* overexpression via *MhcK-Gal4* (Mhc-Gal4.K, BL#55133) suggesting that this Gal4 line is not ideal for uncovering muscle hypertrophy phenotypes.
(TIF)

**S2 Fig. Characterization of the tissue-specificity of transgenic expression with *Mef2-Gal4*.** Red fluorescence due to transgenic *DsRed* expression is detected primarily in body wall skeletal muscles but also in visceral muscles and few cells in the brain. We find no evidence for *Mef2-Gal4*-driven *DsRed* expression in insulin producing cells (ipc) with this line. However, *DsRed* expression driven by *Mef2-Gal4* is detected in brain cells of the mushroom body (mb), consistent with our original characterization of this driver (Demontis and Perrimon, 2009, *Development*; PMID:19211682) and a more recent study that has found endogenous *Mef2* expression in a subset of Kenyon cells of the mushroom body (Crittenden et al. 2018, *Biology Open*; PMID:30115617).
(TIF)

**S1 Table. RNAi screen data.**
(XLSX)

**S2 Table. RNA-seq data.**
(XLSX)

**S3 Table. Phenotypes of RNAi lines for glycolytic enzymes.**
(TIF)

**S4 Table. Additional primary data, related to Figs 2 and 3.**
(XLSX)

## Acknowledgments

We thank the VDRC, the Bloomington stock center, and Dr. Alexey Veraksa for fly stocks. We also thank the Light Microscopy facility and the Hartwell Center for Bioinformatics and Biotechnology at St. Jude Children's Research Hospital. The content is solely the responsibility of the authors and does not necessarily represent the official views of the National Institutes of Health.

## Author Contributions

**Conceptualization:** Flavia A. Graca, Natalie Sheffield, Melissa Puppa, Liam C. Hunt, Fabio Demontis.

**Formal analysis:** Flavia A. Graca, Natalie Sheffield, David Finkelstein, Fabio Demontis.

**Funding acquisition:** Fabio Demontis.

**Investigation:** Flavia A. Graca, Natalie Sheffield, Melissa Puppa, David Finkelstein, Liam C. Hunt, Fabio Demontis.

**Methodology:** Flavia A. Graca, Natalie Sheffield, Melissa Puppa, David Finkelstein, Liam C. Hunt, Fabio Demontis.

**Project administration:** Fabio Demontis.

**Resources:** Flavia A. Graca, Natalie Sheffield, Melissa Puppa, Liam C. Hunt, Fabio Demontis.

**Supervision:** Fabio Demontis.

**Visualization:** Flavia A. Graca, Fabio Demontis.

**Writing – original draft:** Fabio Demontis.

**Writing – review & editing:** Flavia A. Graca, Melissa Puppa, Liam C. Hunt, Fabio Demontis.

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
