## [Decision Letter · Decision Letter 0]

16 Jun 2021

Dear Dr Demontis,

Thank you very much for submitting your Research Article entitled 'A global transgenic RNAi screen identifies transcription factors that modulate myofiber size in Drosophila.' to PLOS Genetics.

The manuscript was fully evaluated at the editorial level and by independent peer reviewers. The reviewers appreciated the attention to an important problem, but raised some substantial concerns about the current manuscript. Based on the reviews, we will not be able to accept this version of the manuscript, but we would be willing to review a much-revised version. We cannot, of course, promise publication at that time.

If you decide to revise the manuscript for further consideration at PLOS Genetics, please aim to resubmit within the next 60 days, unless it will take extra time to address the concerns of the reviewers, in which case we would appreciate an expected resubmission date by email to plosgenetics@plos.org.

[LINK]

We are sorry that we cannot be more positive about your manuscript at this stage. Please do not hesitate to contact us if you have any concerns or questions.

Yours sincerely,

Hongyan Wang, Ph.D.

Associate Editor

PLOS Genetics

Gregory P. Copenhaver

Editor-in-Chief

PLOS Genetics

Reviewer's Responses to Questions

**Comments to the Authors:**

Reviewer #1: The manuscript by Graca et al., titled A global transgenic RNAi screen identifies transcription factors that modulate myofiber size in Drosophila does most of what is suggested by the title. The authors do a fairly large and systematic RNAi screen with the goal of identifying novel transcription factors that regulate muscle growth to better understand the mechanisms that underly muscle atrophy and hypertrophy. In the opinion of this reviewer, the manuscript is exceptionally well written, provides clear and easy to interpret figures, and puts forth a mechanisms by which Glycolysis is regulated for the purpose of muscle growth. Although there are some revisions that are necessary to make this manuscript suitable for publication they are relatively minor in scope.

Questions and Comments

1. I personally would not use the word “global” in the title. As stated, 447 out of 708 transcription factors were investigated. Global to me suggests that all transcription factors were examined. I’d recommend “large-scale” or “systematic”

2. Figure 1 and the associated methods suggest that larvae were just categorized as large, small, mishapen, or normal. Additional details indicating the threshold for variation from control necessary to categorize a larva would be helpful, particularly for readers who may try to follow up on this work.

3. The methods associated with Figure 1 list the control as “such as white RNAi”. If multiple different controls were used they should all be listed along with the rationale for using each.

4. Figure 2 represents my biggest and most important questions regarding this manuscript.

4A. It seems that only a subset of the hits from the screen were examined in figure 2. I certainly recognize that completing this analysis of a large number of different conditions might be prohibitive. However, I think it is essential that the authors provide a rationale for choosing the hits they follow up on and ignoring the hits that they did not. If in fact all of the hits were examined, it should be stated explicitly and data for all should be provided in the supplement at the very least.

4B. The hemisegment chosen for muscle size analysis will impact the data. Therefore it is critical to know which hemisegments were used. Additionally, and perhaps more importantly, the hemisegments near the anterior and posterior end of the animal are significantly smaller and more variable than the central hemisegments. Based on figure 2A it seems that a subset of muscles from these hemisegments may have been used. The authors should remove these data points and rely on the data from central hemisegments.

5. I hesitate to suggest this because it has the feel of making the paper what I would like it to be rather than what it is. Nevertheless, figure 5 just seems an anticlimactic way to end a very nice manuscript. It is suggestive, along with the transcriptomic data in figure 4, that Deaf1 works through regulation of glycolysis to regulate muscle growth. But, we already knew that disruption of glycolysis would impact muscle size. If the glycolysis disruption experiments could be done in the background of altered Deaf1 and/or GSK3 levels, it would be a more fitting conclusion to this manuscript. Even selecting just two glycolytic enzymes to do in this background would be a much more convincing way to show that glycolysis works downstream of Deaf1 and GSK3 to regulate muscle growth.

Minor Comments

1. Citations vary between author, year and numbers. I presume this will be fixed and is not a concern, rather something I noticed.

2. At the bottom of page 6, the authors cite two papers as evidence that fusion is not responsible for the variations in muscle size. I do believe that this is not a fusion effect, but I do not believe that these specific disruptions were looked at in those papers. If I am correct, the authors should be clear and precise in their language.

3. Regarding the statistical analysis, the authors state that all data points indicate a biological replicate. I think that this is vague and means different things to different researchers. For example, in figure 2 one might assume that each point refers to a different mating of males and virgins whereas another might think it means a different animal from a single mating. A more precise description will be helpful in understanding the variability of the data.

4. The last sentence of the first paragraph on page 4 suggests that the sickle shape is indicative of defective contractile properties. Without something to reference, or data to support this conclusion, the authors should just report their finding.

Reviewer #2: The manuscript by Graca et al. reports the results of a straightforward RNAi screen in Drosophila designed to uncover roles for conserved transcription factors in muscle size control. The authors employed 1114 RNAi lines that targeted 447 transcription factors that showed some similarity to human transcription factors, for knockdown in developing muscles and then noted size defects at larval and adult stages as well as some wing positioning defects that have been noted to result from muscle abnormalities. In all about 12% of the targeted genes generated a phenotype consistent with and effect on muscle growth. The authors then performed followed up proof of principle studies on DEAf1 a conserved factor that was not known to affect muscle growth. Loss of Deaf1 was found to increase muscle size while overexpression led to smaller muscles. These phenotypes where opposite to those exhibited by manipulation of GSK3, a kinase that has been shown in mammal system to associate with and phosphorylate DEAf1. Interestingly transcriptome analysis of Deaf1 verses GSK3 knockdown and overexpression n lead to a strikingly similar regulation profile for the core genes involved in glycolysis consistent with pervious observations that inhibition of normal glycolytic flux leads to smaller muscle size.

Overall, this is a short but informative communication that will provide a useful resource to those that study muscle size control in both Drosophila and mammals. In general, the experimental design and outcomes are well described and the results easy to follow. That said I think there are a few points that the authors should address before publication.

1) There should be some discussion of the limitations of the screen. Obviously, there will be some false positives as well as false negatives. It is a bit hard to know what these limits are but one criterion for a true hit might be that more than one RNAi targeting a given gene gives a similar phenotype. By my count there are ~34 genes in supplement table 1 that satisfy this criterion. I think it would be useful to list these genes in a separate table within the main results and provide some limited discussion about these hits such has home many of these have been previously linked to muscle development. Likewise, it is worth pointing out that those genes with only one identified RNAI hit might simply be off target effects. Using this same data set the authors should also take the opportunity to provide an example of another type of possible false positive. I note that trachealess might be one such gene. It satisfies the two-hit criterion, but it is a big surprise, since as far as I am aware, there is no known role for trachealess in muscle development. Rather it seems to be specific for regulating genes involved in the formation of the insect oxygen delivery system. Obviously, if the mef-2 Gal4 driver has some off target expression at some stage in trachea, then it could explain the small larval size phenotype. This brings up my main concern with this study which is that only one “muscle-specific” driver has been used (understandable to keep the work effort down). However, one imposed limitation of this methodology is that the results are only interpretable if the driver expression is tight with respect to the targeted tissue. Mef-2 is pretty good, but there is a big problem with some versions of this driver which is that it is also strongly expressed in the ipc neurons (at least in late third instar larvae) that produce and release several key insulin-like peptides. It is incumbent on the authors to explicitly state which mef2 driver line they are using and confirm by crossing to a UAS reporter that it is indeed muscle specific and does not show expression in other tissues especially the IPCs. Without such confirmation, using mef-2 as the sole driver in the screen could produce false positives by alterations in insulin signaling which is a central regulator of body size through several mechanisms. Obviously, rescreening all hits with an ipc “specific driver” to rule out possible effects on insulin signaling is a large amount of work. However, if the authors Mef2 Gal4 line is expressed in IPCs, then perhaps they could just rescreen the 34 genes that satisfy the more than one RNAi hit criteria described above to determine if any of the phenotypes in this class might be caused by the off target Mef-2 expression.

2) A second issue is that the Deaf and GSK3 work leads to suggesting that they work in the same pathway but in opposite ways to modulate glycolysis. One simple test to determine their order of action is genetic epitasis. If GSK3’s effect on muscle size and glycolytic enzyme expression is primarily through altering Deaf 1 activity, then muscle directed overexpression of GSK3CA together with Deaf1 RNAi should produce a Deaf1 phenotype (higher levels of glycolytic enzyme expression and larger muscle size. Have the authors tried such an experiment?

**Have all data underlying the figures and results presented in the manuscript been provided?**

Reviewer #1: Yes

Reviewer #2: Yes

PLOS authors have the option to publish the peer review history of their article (what does this mean?). If published, this will include your full peer review and any attached files.

Reviewer #1: No

Reviewer #2: No

---

## [Decision Letter · Decision Letter 1]

29 Oct 2021

Dear Dr Demontis,

Thank you very much for submitting your Research Article entitled 'A large-scale transgenic RNAi screen identifies transcription factors that modulate myofiber size in Drosophila.' to PLOS Genetics.

The revised manuscript was fully evaluated at the editorial level and by independent peer reviewers. Both reviewers are generally satisfied with the revision, but reviewer 1 has raised additional minor comments on the manuscript.

We therefore ask you to modify the manuscript according to  review 1's recommendations. 

[LINK]

Yours sincerely,

Hongyan Wang, Ph.D.

Associate Editor

PLOS Genetics

Gregory P. Copenhaver

Editor-in-Chief

PLOS Genetics

Reviewer's Responses to Questions

**Comments to the Authors:**

Reviewer #1: The revised manuscript by Graca et al. addressed my questions and is suitable for publication with the potential of minor modifications to address one question I have.

In the last full paragraph of page 4, the authors compare the data from the Mef2 based RNAi screen to the MhcK based RNAi screen. Specifically, the similarities and the differences are highlighted. However, I question whether strict reliance on the statistical measures is appropriate here. For example, CG6724 is said to reduce larval size in Mef2 but not MhcK. But, the MhcK looks very similar to the Mef2. I do understand that the MhcK data can't be compared to the Mef2 control, but I think that this could be added to the new discussion section on the limitations of the screen. Similarly, Pc and trachealess are listed as "marginally impacted". If the authors are going to stick strictly to statistical measures, they should do so in this case also and list those RNAi lines as having no effect. Finally, the variation in the pdm3 phenotype could also be a point of discussion.

Reviewer #2: The authors have done a very through job of addressing my concerns. I think it is a nice study that will be of interest to many in the muscle biology field.

**Have all data underlying the figures and results presented in the manuscript been provided?**

Reviewer #1: Yes

Reviewer #2: Yes

PLOS authors have the option to publish the peer review history of their article (what does this mean?). If published, this will include your full peer review and any attached files.

Reviewer #1: No

Reviewer #2: No

---

## [Editor Report · Decision Letter 2]

4 Nov 2021

Dear Dr Demontis,

We are pleased to inform you that your manuscript entitled "A large-scale transgenic RNAi screen identifies transcription factors that modulate myofiber size in Drosophila." has been editorially accepted for publication in PLOS Genetics. Congratulations!

Yours sincerely,

Hongyan Wang, Ph.D.

Associate Editor

PLOS Genetics

Gregory P. Copenhaver

Editor-in-Chief

PLOS Genetics

Comments from the reviewers (if applicable):

**Data Deposition**

http://datadryad.org/submit?journalID=pgenetics&manu=PGENETICS-D-21-00604R2

**Press Queries**

---

## [Editor Report · Acceptance letter]

9 Nov 2021

PGENETICS-D-21-00604R2 

A large-scale transgenic RNAi screen identifies transcription factors that modulate myofiber size in Drosophila. 

Dear Dr Demontis, 

We are pleased to inform you that your manuscript entitled "A large-scale transgenic RNAi screen identifies transcription factors that modulate myofiber size in Drosophila." has been formally accepted for publication in PLOS Genetics! Your manuscript is now with our production department and you will be notified of the publication date in due course.

With kind regards,

Zsofia Freund

PLOS Genetics

On behalf of:
